



# Determination of heavy metal fractions in the sediments of oxbow lakes to detect the human impact on the fluvial system (Tisza River, SE Hungary)

M. Tamás, A. Farsang

Department of Physical Geography and Geoinformatics, University of Szeged, POB 653, Szeged H-6701, Hungary

Correspondence to: A. Farsang (farsang@geo.u-szeged.hu)

**Abstract.** Hungary is rich of natural values, but often the condition of them does not meet our expectations. Oxbow lakes and floodplain areas of the South-Plain represent very significant value, not only from landscape, ecological or national tourism point of view, but it is also important from conservation aspect. The scale and

ecological risk of contaminations deducted on rivers that are trapped in floodplains, especially in beds of oxbow lakes, can be judged by examining the sediment of the oxbow lakes. Though data from sludge analysis can provide information about the condition of the oxbow lakes, from which long-term processes or potential ruination of oxbow lakes can be concluded.

In autumn 2011 we have collected samples of sediment from seven oxbow lakes of Lower-Tisza, out of these

four were located in floodplain and three outside the dam. Following the suggested method of Commission of the European Communities Bureau of Reference we analyzed the sample's arsenic, zinc, cadmium, lead, nickel, cobalt, manganese, chromium and copper content with four step BCR sequential extraction, in this way we defined the acid extractable, reducible, oxidisable and residual phases of the heavy metals can be found in the sediment.

It can be stated in all fractions that the contamination matter content in floodplain oxbow lakes is higher than outside the dam. Arsenic, manganese and zinc are present in the sediment of the oxbow lakes in an easily mobilizable form. Regarding the third fraction of arsenic a high metal proportion (15% < III. fraction < 55%) can be observed, whilst examining cadmium and in some cases manganese in specific oxbow lakes a high proportion of fraction one and two can be identified. Lead and chromium are present in a strongly bounded form (IV.

fraction > 90%). During utilization and re-cultivation of oxbow lakes a close attention to be paid to elements in an easily mobilizable form like arsenic, cadmium, zinc and manganese, as these elements can damage the natural vegetation, respectively can get into the food chain through fishing, irrigation or by outplacing sludge onto plough-land.

Keywords: sediment quality of oxbow lakes, sequential extraction, metal contamination





## 1. Introduction

Oxbow lakes of Southern part of Tisza-river in Hungary are evolved by cutting the river-curves or natural unchaining of some reaches during the river regulation in the 19th century. Oxbow lakes of the South-Plain represent very significant value, not only from landscape, ecological or national tourism point of view, but also
from conservation aspect as it assures territory for several rare plant and animal species. Through the river regulations some river-curves got into inside the dams whilst others came to outside the dams, that is how floodplain and outside the dam oxbow rivers issued. Since the two types of the oxbow lakes show large differences in their evolution and also in the effects those are impacted, we handled the two types separately. By analyzing the sediment of the oxbow lakes the accumulation scale of contaminant can be determined, with the
status of the oxbow lakes and the necessity of rehabilitation can be concluded. The definition of the occurrence form of the often very high metal-content is important, since the dredged sludge is often placed out to agricultural soil during the rehabilitation of the oxbow lakes, so the mobilizable element proportion can be a bottleneck of the utilization of dredged sludge, in this way it makes the possibilities of rehabilitation of oxbow lakes more difficult.

The contamination index can be used to systematize the oxbow lakes based on the total heavy metal presence. The contamination index explains for each metal what proportion occurs in the specific oxbow in comparison with the measured highest concentration. With this index we can show the relative contamination ratio of the examined backwaters (Tamás et al., 2011).

Nevertheless the contamination index does not show the ecological and environmental risk of the heavy metal
load. With the Hakanson method the contamination factor and the contamination rate, respectively the status of the sediment and the environment can be evaluated. The method considers the attributes of the heavy metals, so not only the amount of the heavy metals in the sediment can be provided, but its environmental and ecological effect and scale can be concluded (Hakanson, 1980; Spencer, MacLeod 2002; Tamás et al., 2012).

Most analyzes in Hungary were concentrated on the total metal content of the sediment in the floodplain and the
sludge of the oxbow lakes. At the Upper-Tisza region, near the Boroszlókert Holt-Tisza, floodplain sediment was analyzed by Szabó and Posta (2008). The results of the research were evaluated with statistical methods by Szabó et al. (2009). In order to extract the easily mobilizable element proportion of the sediment in Upper-Tisza, Szabó at al. (2008) applied the Lakanen-Erviö extract method. After the pollution of the Upper-Tisza in 2000, floodplain soil contamination was determined by Alapi and Győri (2003). However the heavy metals can stem
from several different sources either natural origin or consequence of human activity. The research results of metal-occurrence can be precise if the natural heavy metals from the geogenic background can be separated from contaminants from anthropogenic sources (Dawson and Macklin, 1998). From this reason sequential extract method was used by Dawson and Macklin (1998) to analyze the floodplain sediment of Aire-valley in Great-Britain or by Birch et al. (1999) in the catchment basin of Parramatta river in Australia.

Therefore it is not sufficient to define the total metal content of the sediments, to determine the origin of the contaminated material and estimate the ecological impact of it. It worth to separate the contaminants and heavy metals based on occurrence (chemical) forms, though the mobilizabilty of the metals and the accessibility for plants can be concluded.

In the last 30 years several extract methods have been worked up to define the metal-occurrence in the
sediments. Usually three-five steps extraction methods are used on sediments. The most often used method's



principles were defined by Tessier et al. (1979) and Kersten and Forstner (1986). The Tessier-method is a five-steps method that in the course of the following metal-fractions can be extracted: 1. exchangeable 2. carbonate bounded 3. Fe and Mn oxide bounded 4. sulphide and organic substances bounded 5. residual.

This relatively simple method has been reworked and improved by several researchers that they modified the reagents used for the extraction (Borovec et al., 1993; Campanella et al., 1995; Zdenek, 1996; Gomez-Ariza et al., 2000; Maiz et al., 2000; Pagnanelli et al., 2004). Therefore several different methods are applied around the world. Towards to eliminate the different extraction procedures and to define a uniform method, the BCR (Community Bureau of Reference) technique (Ure et al., 1993) was created. In 1999 the method has been developed in the frame of European Community (EC) Standard Measurement and Testing Programme (SM&T) (earlier Commission of the European Communities) aiming standardization. The detailed description of the procedure can be found on several places (Rauret et al, 1999; Sahuquillo et al, 1999).

The procedure describes three sequential extraction processes that let the separation of metals into the following forms: 1. acid soluble (Fraction I), 2. reducible (Fraction II), 3. oxidizable (Fraction III). During the definition of the acid soluble phase it defines the exchangeable metal-proportions and the fractions bound to carbonates. The metal-proportions extracted during this solution process can easily get into the waterspout (i.e. in case of pH change) from the sediment. This part of the metal content bound to the sediment the weakest, though these are the most danger to the environment. The second, reducible, phase contains the metal-proportion occurrences that are bind to Fe and Mn oxides. These metal forms are freed up in that case when the sediment changes from oxic to anoxic (i.e. activity of the microorganisms in the sediment). The third, so called oxidizable, phase is that part of the metals that are bound to organic substances respectively to sulphide. These freed up under oxidative circumstances. These conditions can come off in several cases, for example when the sediment re-suspense (in case of dredging, flood or ebb and flow effect) and the particles of the sediment get in contact with oxygen-rich water. After the extraction of the three phases the definition of the forth so called residual phase (Fraction IV) can follow. This fourth phase can be calculated by subtracting the sum of the first three phases from the total metal-content. The metal-content that is present in this phase is bound to the minerals and is component of the crystal structures. This results that there's a high probability that this fraction will not be extracted from the sediment. The standardized BCR extraction method has been used on different areas by several researchers (Togalioglu et al., 2000; Quevauviller et al., 1993; Rauret et al., 1999; Wang et al., 2002; Morillo et al., 2004; Lopez-Sanchez et al., 1996; Kuang-Chung et al., 2001; Yuan et al., 2004; Fuentes et al., 2008; Davidson et al., 1994; Sahuquillo et al., 1999; Salomons 1993; Fiedler et al., 1994; Usero et al., 1998; Martin et al., 1998; Agnieszka and Wieslaw, 2002). The method has been taken over in practice in Hungary and was used during examinations of Balaton and the sediment of the catchment basin of it (Weisz et al., 2000).

Commission of the European Communities defined the BCR CRM 601 method. Compared to the original BCR extraction method the only difference is in the second fraction, namely that way the concentration of the solution ($NH_2OH \cdot HCl$) has been increased and the pH of it has been decreased. This improved the reproducibility thank to it made the extraction of reducible phase of soil/sediment more efficient, though let the Fe-hydroxide phase more available. During the comparison of the original and the modified extraction methods it has been concluded that the modified BCR technology might capable to extract the reducible Fe-based components in soils and sediments (Mossop and Davidson, 2003).



## 2 Studying area

In autumn 2011 we sampled 7 oxbow lakes at Lower-Tisza area that are represented on Figure 1, namely these are Csongrádi-, Osztorai-, Mártélyi-, Körtvélyesi-, Nagyfai-, Atkai-, Saséri oxbow lake. Four of the oxbow lakes are on the floodplain, while three are outside the dam.

*1. Figure The sampled oxbow lakes in the Lower-Tisza region*

The general information of the analyzed oxbow lakes is presented in Table 1 (Pálfai, 2001).

*1. Table General information of the analysed oxbow lakes (Pálfai, 2001)*

The Csongrádi Holt-Tisza oxbow is located on the right side of the river outside the dyke. The oxbow lake was arisen during the cutoff  number 84 in 1860, alluvionation is mediocre, reed and sedge coverage of the surface is 20-25%. Since it does not have a direct connection to the live Tisza the water supply is assured by inland waters, rainwaters and leaking waters  (Pálfai, 2001).

The Osztorai Holt-Tisza oxbow formed on the left side floodplain in 1880 during the regulation of Tisza. Alluvionation of its bed is late in life, coverage by water vegetation is moderate. High level flood waves of Tisza floods the area. It can be drained through the worksless floodplain canal can be found on the lower end. Its function is angler and fisher water, it is a beloved beauty spot. Although its wildlife is diverse and it is reach in animal species, it is currently not a reserved area (Pálfai, 2001).


The Mártélyi Holt-Tisza was arisen during the activities of cutoff number 86 on the left side floodplain of Tisza between 1889 and 1892. The oxbow lake is connected with Tisza through the canal on the lower end. Alluvionation is moderate, water quality is variable, and sometimes signs of eutrophication are shown. The function of the oxbow is storing and carrying irrigation water, which favorable effect on the quality of the water. Beside of this the oxbow lake is used for inland inundation, line-fishing, water sports and the surrounding areas are used for recreation. Its wildlife is reach and various (Pálfai, 2001).


The Körtvélyesi Holt-Tisza was created with the cutoff number 87 in 1889. It is located on the left floodplain of Tisza, coverage by water vegetation is very late in life, alluvionation is mediocre. The oxbow lake is connected with Tisza through the canal on the lower end. Its water supply is assured by rainwaters, leaking waters and flood waves. Its wildlife is very various and rich (Pálfai, 2001).


The Nagyfai Holt-Tisza arose at cutoff number 89 on the left side of Tisza outside the dyke during the regulations started in 1862. Alluvionation is late in life, water vegetation coverage is low, quality of the water is slightly polluted. It is in connection with Tisza through irrigation system. Its functions are storage of inland inundation, irrigation water and used for fishing (Pálfai, 2001).


In 1862 during the regulation of Tisza, at cutoff number 83 an oxbow lakes formed on the right side of the river that was divided into two parts as the result building the dyke, one (smaller) side got on the floodplain while the other to outside the dyke. The first is called Atkai Holt-Tisza the second is the so called Sasér. Both oxbow lakes are nature reserves. (Pálfai, 2001).

The Atkai Holt-Tisza is used for irrigation water storage and line-fishing. Towards to upkeep fishing activities in

the area there are frequent fish-settlements mainly trout, grass carp and silver carp.  The fish-stand of the Atkai





oxbow lake should present the native species; as the settlements of herbivore fishes can be dangerous for the ecological balance of the water. (Kákonyi, 1993). The stock of water of the oxbow lake is assured by rainwater, inland inundation and during high water of Tisza the flood-gate can be found on the southern part of the oxbow lake. Towards to keep the ecological balance it would worth to eliminate the frequent and significant water level

fluctuation. Alluvionation of the oxbow lake is late in life, coverage by water vegetation is slight.

The alluvionation of Sasér is late in life, coverage by water vegetation is little (Pálfai, 2001). Thank to the oxbow lake is located on a highly protected area the plot has a close-to-natural status and it is typified with a various flora and fauna: the waterfront is forested; it is surrounded by willow trees. The oxbow lake's vegetation coverage is exiguous. In the milieu of the oxbow lake there are several rare animal species (Kákonyi, 1993). The

floodplain oxbow lake has constant and direct connection with live Tisza, not only supplied with water through underground leaking, but during flood the pouring Tisza carries significant amount of water to the bed of Sasér, as well as with the water it lays down remarkable amount of sediment in the area of the oxbow lake. Through the floods the water quality of the oxbow lake equals to the water quality of Tisza, as a result of sediment carrier activity of the river the alluvionation of the bed is vigorous. Despite of the frequent water supply, as the oxbow

lake – thank to the regular floods – is in the state of accelerating fill-up, in the last couple of years it dried up several times (Kákonyi, 1993; Pálfai, 2001).

### 3 Methodology

#### 3.1 Sampling and evaluation methods

In each and every oxbow lake we collected sample of the sediment from one area. The samples were always

collected at the inner arc of the oxbow lake, close to the middle of the arc, around 2-3 meters from the waterfront, at roughly 1 meter water depth, from the top 10 cm of the sludge with manual sampler. In average we collected six samples from one square meter area then we homogenized these with blending.

We dried the sludge samples in the laboratory on 105°C temperature for 12 hours and then we milled them in friction-mortar, while we removed the remains of feral and organic residues.

To reduce the inaccuracy of the measure, we executed the steps of the sequential extraction method twice on every sample of the sediment.

The data processing was executed with Microsoft Excel 2003 and SPSS for Windows 16.0. The P-value was evaluated with two-tailed t-test at 95% confidence interval.

We qualified the investigation results based on „B" contamination limit value that can be found in the 6/2009.

(IV.14.) KvVM-EüM-FVM regulation, that defines the permissible maximum concentration of contaminants in geological medium.

#### 3.2 Steps of the sequential extraction

After the preparation as the first step of the extraction was to place 0,8 g of the sludge sample into a 50 ml

centrifugal pipe then we measured 32 ml 0,11 mol $l^{-1}$ acetic acid onto it. The sample was shaken for 16 hours on room temperature. At 4000 rpm we centrifuged the samples for 20 minutes, then the solid residuals have been separated from the solution. We filtered the floating part and stored in plastic container at 4°C temperature. We




washed the residue with 16 ml distilled water with 15 minutes shaking. Then we centrifuged it again and removed the water part.

In the second step we added 32 ml hydroxyl ammonium chloride (0,1 mol l-1 concentration nitric acid adjusted to pH 2) onto the sample left in the centrifuge pipe. This has been shaken for 16 hours on room temperature. We

separated the solution and stored in a plastic contained on 4°C. we washed the residue with distilled water as described in step one.

During the third step of the extraction we added 8 ml hydrogen peroxide (8,8 mol l$^{-1}$) onto the solid-state residue from the second step. We covered the solution with a stopper and stored on room temperature for one hour. Then we uncovered and boiled it in water bath on 85°C for one hour. Then we added again 8 ml hydrogen peroxide

and after covering the centrifuge pipe we boiled it on 85°C for an additional hour.

We added 40 ml ammonium acetate (0,1 mol l-1 concentration nitric acid adjusted to pH 2) onto the cooled residue, then it has been shaken for 16 hours at room temperature. Finally we centrifuged the solution and by filtering we separated from the floating part. The solution is stored in plastic container on 4°C temperature.

Following the steps of sequential extraction method we defined the residual metal-content with aqua regia

extraction method. We flushed the residue of the third step with some ml of distilled water, the dried it. The so got sample was shaken in 10 ml aqua regia for 16 hours the extracted on 180°C. The filtered solution was replenished to 100 ml and stored in plastic containers on 4°C. The theoretical steps of the method are summarized in Table 1.

The heavy metal extraction was executed in the Laboratory of Department of Physical Geography and

Geoinformatics, University of Szeged. The measurement was executed on Perkin Elmer 3110 AAS-flame type equipment following the method defined in the instrument guide.

*Table 2. Used reagents, extraction steps and the phases, occurrence form of the extracted*

## 4 Results

### 4.1 Total metal content in the examined oxbow lakes

The total metal content (Residual phase, Fraction IV) of the oxbow lakes is represented in Table 3. The table shows that the concentration of metals in the floodplain is higher than outside the dyke. Sludge of the floodplain exceeds the 'B' limit value (bold text in the table) regarding the contaminants in geological medium defined in regulation 6/2009. (IV. 14.) KvVM-EüM-FVM more often than the sediments outside the dyke (Tamás and

Farsang, 2011). Our prior measure results show, that nickel, cadmium, zinc and copper content measured in the sediment of the oxbow lakes exceeds the 'B' contamination limit value defined in the regulation several times (Tamás et al., 2011).  The p-values for all measured elements show significant differences between the two type of oxbow lakes (p<0,05) (Tamás and Farsang, 2012).

*Table 3. Measured total heavy metal content in the sediment of oxbow lakes (mg/kg) compared to regulation 6/2009. (IV. 14.) KvVM-EüM-FVM limit values*





### 4.2 Distribution of metal occurrence form in the examined oxbow lakes

Table 4 shows the measured average metal content according to occurrence form in floodplain (n=8) and outside the dyke (n=6). Considering the average values it can be stated that the heavy metal content of the sediment of the oxbow lakes on the floodplain exceeds the outside the dyke in all phases (except Cr-content in Fraction I).

In regard the P-value it can be stated that the difference between floodplain and outside the dyke is significant (Tamás and Farsang, 2012). The insignificant differences of the P-value is marked bold, As and Mn does not show significant differences neither in Fraction I, II, III.

*Table 4. Average and deviation values of heavy metal content in the sediment of oxbow lake based on occurrence form*

On Figure 3 the percentage of occurrence form of the metals in the samples of the sediment is presented. Analyzing the Zinc element all four fractions can be separated. Faction I, II and III of the floodplain oxbow lakes collectively show a value between 40% and 60%, while outside the dyke is between 15% and 30%. The biggest difference between the two oxbow lake types can be identified in the case of Fraction III and IV. Examining

element Zinc it can be concluded that the Zn-content of the sludge outside the dyke can be mobilized harder than in the floodplain.

Regarding the occurrence forms of cadmium it can be concluded that the acid soluble (Fraction I) phase is really high in the case of the Atkai and the Saséri oxbow lakes, it is present around 40% proportion compared to the total cadmium content. Fraction II oscillates between 20% and 30%, while Fraction III and IV changes on a wide

scale. Regarding the occurrence forms of cadmium there is no relevance can be concluded between the floodplain and outside the dyke. Fraction I, II and III of Saséri oxbow lake together exceed 90%, so around 10% of the cadmium-content of the sludge of Saséri oxbow lake is present in bound form.

Great proportion of lead present in bound, hardly mobilizable form in the sludge of the oxbow lakes at Lower-Tisza. Fraction I and II – even if in small proportion – are present in all sludge of the oxbow lakes. Fraction III

was not extractable in the cases of two oxbow lakes (Nagyfai and Atkai), therefore lead is not present in the oxbow lakes bound to organic substances or to sulphide.

Regarding nickel all four fractions can be separated. The highest proportion (80-90%) can be found in Fraction IV videlicet great proportion of the total nickel-content is present in hardly mobilizable form in the analyzed oxbow lakes. Fraction three also represents a significant proportion with its values between 5% and 15%.

Fraction I and II together show a really low proportion, altogether 10% of the total nickel-content. Considering the occurrence forms of nickel in the two types of the oxbow lakes no principals can be determined.

Cobalt in the sludge of Atkai oxbow lake is the least mobilizable (Fraction IV > 90%), while in the case of Sasér oxbow lake Fraction I, II and III together is more than 20%. All four Fractions can be extracted in all examined oxbow lakes however differences in the occurrence of cobalt in two oxbow lake types cannot be defined.

During the analyzes of chromium it can be stated that out of the four examined Fractions the occurrence of Fraction II is present in a negligible amount (max measured value is 0,2086 μg/kg) just like Fraction I (measured values between 0,042175 and 0,25868 μg/kg) videlicet chromium is present bound to Fe and Mn oxides respectively to organic substances and to sulphide in a very low scale. Fraction III pulses between 5% and 10% in the sludge of the analyzed oxbow lakes, but due to the low values of Fraction I and II the first three fractions

together does not exceed 15%.





Regarding the occurrence forms of copper it can be concluded that Fraction II is present in a really low proportion (<5%). Except Csongrádi oxbow lake Fraction I never exceeds 5%, while Fraction III in all oxbow lakes oscillates between 5% and 25%. Great proportion (~20%) of the copper-content can be easily mobilized in all of the examined floodplain oxbow lakes, oddly considering the high proportion of Fraction III (>10%).

Regarding the occurrence forms of copper in the two oxbow lakes no principals can be concluded.

Analyzing the occurrence forms of arsenic all four Fractions can be extracted. It can be stated that Fraction III is present in very high proportion, the oxidizable phase is between 15% and 55% therefore the hardly mobilizable proportion never exceeds 85% in the examined oxbow lakes. In the case of Csongrádi oxbow lake Fraction IV does not reach 40% and Fraction III exceeds 50%.

The manganese-occurrences show different results, than the previously observed metals, oddly considering the high proportion of Fraction I and II. Regarding the occurrences of manganese it can be seen that it can be easily mobilized, namely the proportion of Fraction I exceeds 20% in all of the examined oxbow lakes. In the case of Csongrádi oxbow lakes the proportion of Fraction I is extremely high, more than 40%. Fraction I, II and III altogether exceeds 40% in all oxbow lakes, and it is raised high in the case of Nagyfai oxbow lake, where this

amount is ~85%.

*Figure 2. Distribution of occurrence forms of metal sin the oxbow lakes*

**5 Disputation**

Separating the elements measure in the sludge of the floodplain and outside the dyke oxbow lakes we defined a

mobility order based on the proportion of total metal-content and mobilizable fraction. The mobility order of the sludge of floodplain oxbow lakes is the following: Pb<Cr<Ni<Co<As<Cu<Zn<Mn<Cd, while outside the dyke it is: Pb<Cr<Cu<Co<Ni<Zn<As<Cd<Mn. Therefore it can be concluded in both examined areas that the least mobilizable elements are Pb and Cr, while the "most critical" are Cd and Mn.

Comparing our examination results to the results of earlier executed Hungarian sludge-analyzes of the sediment

of Balaton it can be concluded that except cadmium the examined elements (zinc, nickel, chromium, copper, lead) is present in more bound form in the sediment of the oxbow lakes in the Tisza-region than in the sediment of Balaton (Bódog et al., 1996). The seemingly high proportion of easily mobilizable cadmium is not unusual as other explorations show similarly high measures in regard of Fraction I, II and III (Morillo et al., 2003; Fuentes et al., 2007).

In regard of manganese Tokalioglu et al. (2000) measured similar proportions using three step BCR technology during his researches in Sultansazligi, Kayseri in Turkey, like the proportions can be identified on the sediment of the oxbow lakes at Tisza. The easily mobilizable manganese was also present in the Estonian oostrikui turf bog (Syrovetnik et al., 2005).

Box-plot diagrams on Figure 3 show by Fractions the measured metal-concentration in the sludge of outside the

dyke (n=3) and floodplain (n=4) oxbow lakes. It can be read from the diagrams (and we verified with calculations) that in the cases of zinc, cadmium, lead, nickel, cobalt, chromium and copper considering the medians, examining the four Fractions the metal-concentration in the floodplain oxbow lakes is higher than the outside the dyke ones.





Analyzing Fraction IV it can be clearly seen on the diagrams that in regard the median the floodplain oxbow lakes show higher load than outside the dyke oxbow lake- quadrants (except manganese-concentration). The lower- and upper-quartiles can be seen on the diagram, furthermore I can be identified that there are no raised values in regard any item of the sample-set.

*Figure 3. Metal-concentration of samples of floodplain and outside the dyke by Fractions*

Analyzing the distribution of elements based on Fractions, that from total content perspective exceeds the contamination limit value defined in the regulation it can be stated that in the sludge of floodplain and also outside the dyke oxbow lakes nickel (80%, respectively 84%) and chromium (90%, respectively 94%) is present in a hard-bound form at a very high proportion. High proportion (78%) of the zinc in the sediment of outside the dyke oxbow lakes is present in a hard-bound form, while this proportion is just 49% in the sludge of floodplain oxbow lakes. In the case of cadmium it can also be stated that it is present in higher proportion (42%) in hard-bound form in outside the dyke oxbow lakes, than in the sludge of floodplain (32%). While in regard of the total element-content we would place nickel, chromium, zinc and cadmium into focus to qualify the sludge of the oxbow lakes, respectively during the analyzes of potential ecological impact of the sludge, then until it can be concluded that – as those are present in hard-bounded form – nickel and chromium is "not critical" element. However we have to consider the zinc-content and cadmium occurrences when the sludge is placed on plow-land or the ecological status of the oxbow lakes has to be defined, also like manganese and arsenic elements that are present in a hard-bound form in the sludge in less than one third of the total content.

## 5 Summary

The average values of the nickel-, cadmium-, zinc-, and copper-content of the sediments in several floodplain oxbow lakes exceeds the „B" contamination limit value that is defined in the 6/2009. (IV.14.) KvVM-EüM-FVM regulation, that defines the permissible maximum concentration of contaminants in geological medium. The contaminant content of the sediment in the oxbow lakes outside the dyke never exceeded the limit value defined in the regulation. Higher contaminant concentration can be identified in the floodplain oxbow lakes then oxbow lakes outside the dyke. This relation is true in the case of zinc, cadmium, lead, nickel, cobalt, chromium and copper elements in all four Fractions.

Based on the results of the sequential extraction it can be concluded that towards the surveying of the status of the oxbow lakes, respectively the definition of the potential recultivation it is not satisfactory to examine the total contaminant matter of the sediment. Based on the total contaminant content solely the zinc-, cadmium-, nickel- and copper-content can be a bottle-neck of the rehabilitation of a floodplain oxbow lake. Nickel and copper element are present in hard-bounded form in the sediment of the oxbow lakes (10% > Fraction I+II), so these elements does aggravate the allocation of the sediment. Based on the results of the sequential extraction for the recultivation of an oxbow lake it is necessary to examine the contaminant content of the sediment of both floodplain and outside the dyke oxbow lakes, oddly attention to arsenic-, cadmium-, zinc-, and manganese-





content, since these are present in the oxbow lakes in an easily mobilizable form  in the sediment of the oxbow lakes.

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

**Figures:**

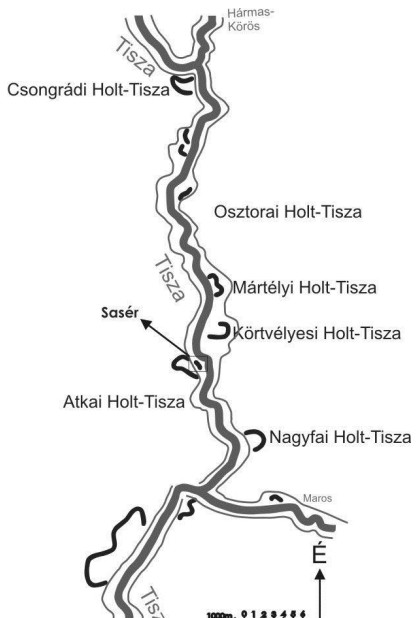

Figure 1. The sampled oxbow lakes in the Lower-Tisza region





**Figure 2. Distribution of occurrence forms of metal sin the oxbow lakes**



**Figure 3. Metal-concentration of samples of floodplain and outside the dyke by Fractions**





**Tables:**

| Name of oxbow lake | Length (km) | Average width (m) | Surface (ha) | Average depth (m) | Water volume (m³) | Reserve degree | Formation date | Location |
|---|---|---|---|---|---|---|---|---|
| Csongrádi | 7,5 | 174 | 130 | 2 | 2,6 million | Not reserved | 1860 | Outside the dam |
| Osztorai | 2,5 | 100 | 25 | 1,6 | 400 thousand | Not reserved | 1880 | Floodplain |
| Mártélyi | 4,6 | 100 | 46 | 2 | 920 thousand | Nature reserve | 1889 | Floodplain |
| Körtvélyes | 4,7 | 128 | 60 | 3 | 1,8 million | Nature reserve | 1889 | Floodplain |
| Nagyfai | 5,9 | 103 | 61 | 1,5 | 915 thousand | Not reserved | 1887 | Outside the dam |
| Atkai | 6,8 | 122 | 83 | 3,5 | 2,9 million | Nature reserve | 1889 | Outside the dam |
| Sasér | 0,97 | 103 | 10 | 1 | 300 thousand | Nature reserve | 1889 | Floodplain |

**Table 1. General information of the analysed oxbow lakes (Pálfai, 2001)**

| Steps | Reagents and extraction method | Fractions | Extracted occurrence form |
|---|---|---|---|
| 1 | 32 ml acetic acid $CH_3COOH$ (0,11 mol $l^{-1}$) pH 2,85, 16 hours | Acid soluble (Fraction I) | exchangeable metal-proportions and the fractions bound to carbonates |
| 2 | 32 ml hydroxyl ammonium chloride $NH_2OH \cdot HCl$ (0,1 mol $l^{-1}$) pH 2, 16 hours | Reducible (Fraction II) | bind to Fe and Mn oxides |
| 3 | 2 X 8 ml hydrogen peroxide $H_2O_2$ 9,8 mol $l^{-1}$ + ammonium acetate $CH_3COONH_4$ (1 mol $l^{-1}$) pH 2; 2 X 85°C-on 2 X 1 hour water bath | Oxidizable (Fraction III) | bound to organic substances respectively to sulphide |
| 4 | 10 ml aqua regia $3HCl+HNO_3$ 16 hours 180°C | Residual (Fraction IV) | subtract the sum of the first three phases from the total metal-content; bound to the minerals and is component of the crystal structures |

5    **Table 2. Used reagents, extraction steps and the phases, occurrence form of the extracted**

| Parameter | Floodplain | | | | Outside the dyke | | | Limit value in geological medium |
|---|---|---|---|---|---|---|---|---|
| | Osztorai | Mártély | Körtvélyes | Sasér | Atka | Nagyfa | Csongrád | |
| As | 14,70 | 13,76 | 8,17 | 11,07 | 7,55 | 9,37 | 5,98 | 15 |
| Zn | 171,37 | **267,06** | 73,33 | **307,03** | 81,08 | 103,38 | 28,73 | 200 |
| Cd | 0,80 | **1,13** | 0,48 | **1,52** | 0,60 | 0,62 | 0,24 | 1 |
| Pb | 42,19 | 60,16 | 11,63 | 39,92 | 19,27 | 23,88 | 6,01 | 100 |
| Ni | **73,48** | **77,79** | 32,13 | **66,90** | 35,00 | **47,88** | 14,07 | 40 |
| Co | 29,31 | 27,69 | 15,73 | 26,74 | 15,82 | 19,80 | 9,44 | 30 |
| Mn | 446,87 | 392,27 | 367,33 | 401,36 | 448,13 | 540,81 | 421,59 | - |
| Cr | **80,09** | **88,57** | 35,53 | 73,93 | 34,49 | 53,21 | 16,63 | 75 |
| Cu | 42,31 | 68,58 | 22,33 | 67,07 | 18,96 | 28,45 | 5,26 | 75 |

**Table 3. Measured total heavy metal content in the sediment of oxbow lakes (mg/kg) compared to regulation 6/2009. (IV. 14.) KvVM-EüM-FVM limit values**





| Examined metal | Fraction I | | | Fraction II | | | Fraction III | | | Fraction IV | | |
|---|---|---|---|---|---|---|---|---|---|---|---|---|
| | Floodplain (n=4) | Outside the dyke (n=3) | P-value | Floodplain (n=4) | Outside the dyke (n=3) | P-value | Floodplain (n=4) | Outside the dyke (n=3) | P-value | Floodplain (n=4) | Outside the dyke (n=3) | P-value |
| As | 0.49 ± 0.05 | 0.48 ± 0.03 | **0,474** | 0.21 ± 0.02 | 0.18 ± 0.07 | **0,283** | 2.08 ± 0.09 | 2.68 ± 0.61 | **0,063** | 11.92 ± 2.78 | 7.63 ± 1.59 | 0,004 |
| Zn | 28.69 ± 26.14 | 5.3 ± 2.85 | 0,040 | 32.77 ± 26.1 | 5.71 ± 4.34 | 0,021 | 42.24 ± 26.1 | 4.36 ± 3.29 | 0,005 | 204.69 ± 98.1 | 71.06 ± 34.6 | 0,006 |
| Cd | 0.24 ± 0.23 | 0.15 ± 0.15 | **0,397** | 0.3 ± 0.2 | 0.1 ± 0.06 | 0,032 | 0.14 ± 0.09 | 0.03 ± 0.01 | 0,012 | 0.98 ± 0.42 | 0.49 ± 0.19 | 0,015 |
| Pb | 0.66 ± 0.38 | 0.43 ± 0.31 | **0,226** | 1.3 ± 0.75 | 0.35 ± 0.21 | 0,009 | 1.29 ± 0.96 | 0.02 ± 0.02 | 0,007 | 38.47 ± 18.7 | 16.39 ± 8.35 | 0,007 |
| Ni | 2.35 ± 0.85 | 1.04 ± 0.54 | 0,004 | 2.94 ± 1.5 | 1.11 ± 0.95 | 0,017 | 6.93 ± 1.43 | 3.08 ± 2.48 | 0,011 | 62.58 ± 19.39 | 32.32 ± 15.7 | 0,007 |
| Co | 1.54 ± 0.69 | 0.64 ± 0.25 | 0,008 | 1.79 ± 1.08 | 0.75 ± 0.88 | 0,069 | 2.04 ± 0.56 | 0.84 ± 0.38 | 0,000 | 24.87 ± 5.92 | 15.02 ± 4.86 | 0,005 |
| Mn | 149.15 ± 26.2 | 183.6 ± 94.3 | **0,417** | 79.33 ± 40.1 | 90.4 ± 96.3 | **0,800** | 38.4 ± 13.22 | 26.6 ± 18.7 | **0,222** | 401.9 ± 35.04 | 470.2 ± 63.6 | 0,049 |
| Cr | 0.12 ± 0.04 | 0.16 ± 0.16 | **0,592** | 0.15 ± 0.06 | 0.07 ± 0.02 | 0,005 | 6.34 ± 2.85 | 1.95 ± 1.09 | 0,003 | 69.53 ± 21.94 | 34.77 ± 16.6 | 0,006 |
| Cu | 2.31 ± 1.42 | 0.75 ± 0.25 | 0,016 | 0.49 ± 0.56 | 0.16 ± 0.04 | **0,143** | 10.36 ± 6.17 | 1.28 ± 0.39 | 0,004 | 50.07 ± 20.5 | 17.56 ± 10.6 | 0,003 |

**Table 4. Average and deviation values of heavy metal content in the sediment of oxbow lake based on occurrence form**