# Peer review of "Determination of heavy metal fractions in the sediments of oxbow lakes to detect the human impact on the fluvial system (Tisza River, SE Hungary)"

_Hydrology and Earth System Sciences, 2016_

## Referee Comment (RC1) · Anonymous Referee #1 · 9 Aug 2016

This manuscript could be of interest to researchers within the field of environmental chemistry. The authors present a comparison of metal contamination of sediments in two types of oxbow lakes, with a detailed description of the sequential extraction methodology adopted in the analyses. However, I have some serious reservations of suggesting publication in its present form. There are two inherent weaknesses in this manuscript. First, the paper focuses too much on presenting the analytical results. There is almost no discussion about the environmental implications of the findings. This limited approach is also a major drawback due to the relative importance of some biogeochemical processes, which are not taken into account in this work. Second,

there are several language issues throughout the text, making it sometimes difficult to understand what the authors are attempting to say. There are also several spelling errors. The paper would thus benefit from a thorough revision.

There are also problems with the structure of the paper. The objectives are not clearly stated. Methods are described in the introduction and new results are presented in the discussion section.

Other minor problems include:

Abstract:

-lack of definition of BCR when it first appears.

Introduction:

-the description of oxbow lakes is not very clear, especially the ones "inside the dams", it took me a while to understand what were the dams;

-references are sometimes displayed incorrectly in the body of the text and do not correspond to what is listed in the respective section (ex. Tamás et al., 2012, instead of Tamás and Farsang, 2012).

Study area:

- for those unfamiliar with the study region, some descriptions are not very clear, for example, "cutoff number 84 in 1860"; the authors should provide some explanation;

-a description of Lower-Tisza area characteristics that could influence the findings should be presented, including factors leading to the contamination of the oxbow lakes;

- Figure 1, which illustrates the study area, needs to be improved. It lacks an overall reference of the study area location, a clear scale, and a legend indicating what the lines represent.

Results:
-results in paragraph 1 are associated to references of other studies. The results presented are not original? Or what is the innovative aspect of the work under review? Discussion:

-as mentioned before, in the discussion section, processes that can influence the observed results and differences between oxbow lake types are very poorly explored.

---

## Referee Comment (RC2) · Anonymous Referee #2 · 26 Aug 2016

The topic of the paper is relevant for HESS. The title is not clearly reflecting to the contents: Title refers to the whole fluvial system of the River Tisza while paper focuses on the oxbow lakes in the Hungarian section of the river.

This paper provides new data about oxbow lakes of the River Tisza. Applied techniques are include standard procedures.

Formulae, symbols, abbreviations, and units are mostly correctly defined and used. (Exceptions are indicated in the pdf) Number and quality of references is appropriate. Amount and quality of supplementary materials are appropriate. Langauge is not fluent

and precise.

Paper introduces a nice study. Authors have significantly new results but the conclusion is missing. Present manuscript should be complemented with a conclusion.

The aim and the assumption of the research is not clearly outlined. However these can be found in the text. Sampling strategy and sampling techniques are adequate. Analytical methods are conventional. Mathematical method is appropriate. Authors use a parameter (B value) of a Hungarian regulation legal rule (6/2009 KvVM-EüM-FVM) as a standard (5 page, line30). This kinds of measures are a compromise. This rule and value is primarily serves the activities of authorities and the administration but it is not for science. The application of the geochemical background would be more informative. Several papers published about determination of geochemical background of metals.

Results of the paper deepen knowledge about the status of the degree of (heavy) metal pollution in oxbow lakes along the River Tisza. But some of the observations are not explained and discussed. E.g.: page 8 lines 19-29: Observations: difference in mobility order in sediments between two kinds of oxbow lakes. Explanation of the phenomenon is missing.

The description of the methods and the scientific work is mostly complete and precise. Description of sampling is correct. Section of lab methods is more detailed than the required. Data processing is acceptable. The sequence of the methods is not appropriate (see in the pdf).

The paper is long-winded. There are some confusion within the chapters (e.g: sequence of methods) and some results appear in the discussion (page 8 lines 19-29).

Introduction is too long. Several lines deal with the history of sequential extraction methods. I recommend to eliminate: page 2. Lines 39-40, Page 3 lines 1-26.

Please also note the supplement to this comment:
http://www.hydrol-earth-syst-sci-discuss.net/hess-2016-207/hess-2016-207-RC2-supplement.pdf

———————————————————————
[Figure]

**Supplement:**

[revised manuscript text omitted]

---

## Author Comment (AC1) · 29 Sep 2016

Answer to comment on "Determination of heavy metal fractions in the sediments of oxbow lakes to detect the human impact on the fluvial system (Tisza River, SE Hungary)" by M. Tamás and A. Farsang

Answer to Anonymous Referee #1

Thank you for the Referees, that they found our article worthy to be considered amongst researchers interested in environmental chemistry.

"The paper focuses too much on presenting the analytical results. There is almost no discussion about the environmental implications of the findings." – We do not wish to shorten the analytical results, but we agree that the discussion, seeing that the visualization of the results of environmental impacts to be strengthen and discussion part to be amended. "There are several language issues throughout the text, making it sometimes difficult to understand what the authors are attempting to say. There are also several spelling errors. The paper would thus benefit from a thorough revision."- The linguistic issues will be corrected and reviewed by a native speaker. "There are also problems with the structure of the paper. The objectives are not clearly stated. Methods are described in the introduction and new results are presented in the discussion section." – We will review the structure of the article and we will apply the re-structuring suggested by the Referee. Other minor problems include: "Abstract: lack of definition of BCR when it first appears." – We will cite the definition. "Introduction: the description of oxbow lakes is not very clear, especially the ones "inside the dams", it took me a while to understand what were the dams" – We will fix the definition. "Introduction: references are sometimes displayed incorrectly in the body of the text and do not correspond to what is listed in the respective section (ex. Tamás et al., 2012, instead of Tamás and Farsang, 2012)" – We will fix it. "Study area: for those unfamiliar with the study region, some descriptions are not very clear, for example, "cutoff number 84 in 1860"; the authors should provide some explanation" – We will provide the necessary explanations. "Study area: a description of Lower-Tisza area characteristics that could influence the findings should be presented, including factors leading to the contamination of the oxbow lakes" – The description of Lower-Tisza area characteristics that could influence the findings will be amended to the article. "Figure 1, which illustrates the study area, needs to be improved. It lacks an overall reference of the study area location, a clear scale, and a legend indicating what the lines represent." – Correction of Figure 1 will be applied. "Results: Discussion paper: results in paragraph 1 are associated to references of other studies. The results presented are not original? Or what is the innovative aspect of the work under review?" – Some small part of the result

has already published, because the new result on built on previous results. "Results: Discussion: as mentioned before, in the discussion section, processes that can influence the observed results and differences between oxbow lake types are very poorly explored." – We will amend the implications and conclusions of the article.

Thanks for the Referees for their suggestions to improve our article.

Szeged, 2016. September 27.

Regards:

Andrea Farsang Dr. Associate Professor University of Szeged Department of Physical Geography and Geoinformatics

Margit Tamás PhD student University of Szeged Department of Physical Geography and Geoinformatics

---

## Author Comment (AC2) · 29 Sep 2016

Answer to comment on "Determination of heavy metal fractions in the sediments of oxbow lakes to detect the human impact on the fluvial system (Tisza River, SE Hungary)" by M. Tamás and A. Farsang

Answer to Anonymous Referee #2

Thanks for the Referee's suggestions that aim to improve the quality of the article. Thank you for judging the article to be considerable for HESS.

"The title is not clearly reflecting to the contents: Title refers to the whole fluvial system of the River Tisza while paper focuses on the oxbow lakes in the Hungarian section of the river." – Thank you for the suggestion in regards the title of the article, we will consider the modification. "Langauge is not fluent." – We will review and correct the article from linguistic point of view and will send for a proof reading. "Authors have significantly new results but the conclusion is missing. Present manuscript should be complemented with a conclusion." – We will amend the implications and conclusions of the article. "Authors use a parameter (B value) of a Hungarian regulation legal rule (6/2009 KvVM-EüMFVM) as a standard (5 page, line30). This kinds of measures are a compromise." – We accept the statement of the Referee that we compared our result against the geochemical background instead of the B value used in the Hungarian regulation legal rule. "Some of the observations are not explained and discussed. E.g.: page 8 lines 19-29: Observations: difference in mobility order in sediments between two kinds of oxbow lakes. Explanation of the phenomenon is missing." – We will amend the necessary explanation. "There are some confusion within the chapters (e.g: sequence of methods) and some results appear in the discussion (page 8 lines 19-29)." – We fix it. "Introduction is too long. Several lines deal with the history of sequential extraction methods. I recommend to eliminate: page 2. Lines 39-40, Page 3 lines 1-26." – We will consider the suggestion to shorten the article, thank you. "Formulae, symbols, abbreviations, and units are mostly correctly defined and used.(Exceptions are indicated in the pdf)" – We accept the suggestions in regards of the formulas, symbols, abbreviations and we will consider to shorten the article. We will amend the missing references.

Thanks for the Referees for their suggestions to improve our article.

Szeged, 2016. September 27. Regards:

Andrea Farsang Dr. Associate Professor University of Szeged Department of Physical Geography and Geoinformatics
Margit Tamás PhD student University of Szeged Department of Physical Geography and Geoinformatics

---

## Author Comment (AC3) · 17 Nov 2016

Answer to comment on "Determination of heavy metal fractions in the sediments of oxbow lakes to detect the human impact on the fluvial system (Tisza River, SE Hungary)" by M. Tamás and A. Farsang

Answer to Anonymous Referee #1 Thank you for the Referees, that they found our article worthy to be considered amongst researchers interested in environmental chemistry. "The paper focuses too much on presenting the analytical results. There is almost no discussion about the environmental implications of the findings." –

We shortened the analytical methods significantly and the analytical results slightly. The discussion part has been amended by focusing on the elements' environmental impact. "There are several language issues throughout the text, making it sometimes difficult to understand what the authors are attempting to say. There are also several spelling errors. The paper would thus benefit from a thorough revision."- The linguistic issues corrected and reviewed by a native speaker. "There are also problems with the structure of the paper. The objectives are not clearly stated. Methods are described in the introduction and new results are presented in the discussion section." – We reviewed the structure of the article and we applied the re-structuring suggested by the Referee. Other minor problems include: "Abstract: lack of definition of BCR when it first appears." – We cited the definition. "Introduction: the description of oxbow lakes is not very clear, especially the ones "inside the dams", it took me a while to understand what were the dams" – Clarified in the introduction. "Introduction: references are sometimes displayed incorrectly in the body of the text and do not correspond to what is listed in the respective section (ex. Tamás et al., 2012, instead of Tamás and Farsang, 2012)" – We fixed it. "Study area: for those unfamiliar with the study region, some descriptions are not very clear, for example, "cutoff number 84 in 1860"; the authors should provide some explanation" – Due to shorten and re-wording this part was reconsidered. "Study area: a description of Lower-Tisza area characteristics that could influence the findings should be presented, including factors leading to the contamination of the oxbow lakes" – We added some information about the main impacts of oxbow lakes depending on the characteristics of Lower-Tisza valley. "Figure 1, which illustrates the study area, needs to be improved. It lacks an overall reference of the study area location, a clear scale, and a legend indicating what the lines represent." – We corrected of Figure 1. "Results: Discussion paper: results in paragraph 1 are associated to references of other studies. The results presented are not original? Or what is the innovative aspect of the work under review?" – Some small part of the result has already published, because the new result on built on previous results. "Results: Discussion: as mentioned before, in the discussion section,

processes that can influence the observed results and differences between oxbow lake types are very poorly explored." – We amended the implications and conclusions of the article. Thanks for the Referees for their suggestions to improve our article. Szeged, 2016. November 17. Regards: Andrea Farsang Dr. Associate Professor University of Szeged Department of Physical Geography and Geoinformatics Margit Tamás PhD student University of Szeged Department of Physical Geography and Geoinformatics

Please also note the supplement to this comment:
http://www.hydrol-earth-syst-sci-discuss.net/hess-2016-207/hess-2016-207-AC3-supplement.pdf

---

## Author Comment (AC4) · 17 Nov 2016

Answer to comment on "Determination of heavy metal fractions in the sediments of oxbow lakes to detect the human impact on the fluvial system (Tisza River, SE Hungary)" by M. Tamás and A. Farsang

Answer to Anonymous Referee #2

Thanks for the Referee's suggestions that aim to improve the quality of the article. Thank you for judging the article to be considerable for HESS. "The title is not clearly reflecting to the contents: Title refers to the whole fluvial system of the River Tisza

while paper focuses on the oxbow lakes in the Hungarian section of the river." – We modified the title of article. "Langauge is not fluent." – We reviewed and corrected the article from linguistic point of view. "Authors have significantly new results but the conclusion is missing. Present manuscript should be complemented with a conclusion." – We amended the implications and conclusions of the article. "Authors use a parameter (B value) of a Hungarian regulation legal rule (6/2009 KvVM-EüMFVM) as a standard (5 page, line30). This kinds of measures are a compromise." – We accept the statement of the Referee that we compared our result against the geochemical background instead of the B value used in the Hungarian regulation legal rule. We amended the Hungarian geological background values. "Some of the observations are not explained and discussed. E.g.: page 8 lines 19-29: Observations: difference in mobility order in sediments between two kinds of oxbow lakes. Explanation of the phenomenon is missing." – We amended the necessary explanation. "There are some confusion within the chapters (e.g: sequence of methods) and some results appear in the discussion (page 8 lines 19-29)." – We fixed it. "Introduction is too long. Several lines deal with the history of sequential extraction methods. I recommend to eliminate: page 2. Lines 39-40, Page 3 lines 1-26." – We shortened the article. "Formulae, symbols, abbreviations, and units are mostly correctly defined and used. (Exceptions are indicated in the pdf)" – We corrected the suggestions in regards of the formulas, symbols, abbreviations and we shortened the article in some cases. We amended the missing references.

Thanks for the Referees for their suggestions to improve our article.

Szeged, 2016. November 17. Regards: Andrea Farsang Dr. Associate Professor University of Szeged Department of Physical Geography and Geoinformatics Margit Tamás PhD student University of Szeged Department of Physical Geography and Geoinformatics

Please also note the supplement to this comment:
http://www.hydrol-earth-syst-sci-discuss.net/hess-2016-207/hess-2016-207-AC4-

supplement.pdf